# Climate Change May Restrict the Predation Efficiency of *Mesocyclops aspericornis* (Copepoda: Cyclopidae) on *Aedes aegypti* (Diptera: Culicidae) Larvae

**DOI:** 10.3390/insects11050307

**Published:** 2020-05-14

**Authors:** Nobuko Tuno, Tran Vu Phong, Masahiro Takagi

**Affiliations:** 1Graduate School of Natural Science and Technology, Kanazawa University, Kanazawa 920-1192, Japan; 2Institute of Tropical Medicine, Nagasaki University, Nagasaki 852-8523, Japan; tranvuphong@yahoo.com (T.V.P.); mstakagi@nagasaki-u.ac.jp (M.T.); 3Department of Medical Entomology and Zoology, National Institute of Hygiene and Epidemiology, Hanoi 100000, Vietnam

**Keywords:** *Aedes aegypti*, biological control, copepod, dengue fever

## Abstract

(1) Dengue is the most spread mosquito-borne viral disease in the world, and vector control is the only available means to suppress its prevalence, since no effective treatment or vaccine has been developed. A biological control program using copepods that feed on mosquito larvae has been practiced in Vietnam and some other countries, but the application of copepods was not always successful. (2) To understand why the utility of copepods varies, we evaluated the predation efficiency of a copepod species (*Mesocyclops aspericornis*) on a vector species (*Aedes aegypti*) by laboratory experiments under different temperatures, nutrition and prey-density conditions. (3) We found that copepod predation reduced intraspecific competition among *Aedes* larvae and then shortened the survivor’s aquatic life and increased their pupal weight. In addition, the predatory efficiency of copepods was reduced at high temperatures. Furthermore, performance of copepod offspring fell when the density of mosquito larvae was high, probably because mosquito larvae had adverse effects on copepod growth through competition for food resources. (4) These results suggest that the increase in mosquitoes will not be suppressed solely by the application of copepods if the density of mosquito larvae is high or ambient temperature is high. We need to consider additional control methods in order to maintain the efficiency of copepods to suppress mosquito increase.

## 1. Introduction

*Aedes* mosquitoes transmit dengue virus that cause dengue fever and dengue haemorrhagic fever [1]. Because no effective vaccine has been developed against dengue virus, control programs of dengue generally target the major vectors *Aedes* (*Stegomyia*) *aegypti* (Linnaeus) and *Aedes* (*Stegomyia*) *albopictus* (Skuse). A noticeable means for *Aedes* control is the use of copepod predators. Since the discovery of copepod predation on mosquito larvae, they have been used to control *Aedes* larvae in temperate, subtropical and tropical regions [2,3,4,5,6,7,8,9]. So far, copepod application has been proven to suppress mosquito populations in northern and central Vietnam [2,3,4,5,10], but copepod application alone was not effective in southern Vietnam [11]. In a review of several copepod application programs, it is claimed that the effectiveness of copepod application varies according to the differences in community structures and environmental conditions [12]. However, still, there have been few experimental studies on this issue [13].

Copepods predate on younger (mainly first instar) mosquito larvae more efficiently in comparison with later-stage larvae [6,14]. Therefore, the effectiveness of copepods to suppress mosquito increase may be reduced at high temperatures or under rich resource conditions, because mosquito larvae develop faster under such conditions and have shorter vulnerable periods. The effectiveness of copepods would also be reduced if food resources other than mosquito larvae are available. In addition, their predation efficiency is reduced in the presence of detrital substrates [15], possibly because such substrates provide refuge for mosquito larvae. Furthermore, mosquito larvae are able to compete with copepods for food when they grow larger [16]. Competitive interactions would lower survivorship and fecundity and prolong developmental time both in copepods and mosquitoes [15,17,18,19,20], and the outcome of their competition would differ if environmental conditions change. To evaluate the effectiveness of copepods on the control of mosquito populations, therefore, it is important to understand how environmental conditions affect copepod’s predatory efficiency and their competitive interactions with mosquitoes. These knowledges are also important to understand the effect of global warming on future maps of arthropod-borne disease [21]. In general, organisms, including *Aedes*, increase more rapidly at a higher temperature if it does not exceed harmful upper limits and food is not limited [22]. However, global warming would not always lead to population expansion if predator activity increases in parallel.

This study aims to understand the effects of environmental conditions on the effectiveness of a copepod species *Mesocyclops aspericornis* (Daday) as an agent to suppress the increase in *Ae. aegypti* populations by laboratory experiments in which rearing conditions, i.e., temperature, the amount of foods for mosquitoes and copepods, and the initial density of mosquito larvae, are manipulated. We used *Ae. aegypti* but not *Ae. albopictus* as a prey species because of the following two reasons. First, the former species is more sensitive to environmental changes in population growth than the latter [22]. Second, when their eggs are soaked in water, *Ae. ageypti* eggs immediately hatch, but *Ae. albopictus* eggs vary in hatching time. Therefore, it is difficult to prepare a large number of first instar larvae for experimental use in the latter.

## 2. Materials and Methods 

### 2.1. Copepods

Mass cultures of *M. aspericornis* were established with females collected from water containers placed in a domestic area of Ho Chi Minh City (10˚46’10”N and 106˚40’55”E), Vietnam in 2003. They were maintained in 15-L plastic boxes (28 × 38 × 14 cm) according to the methods described by Suárez et al. (1992). Copepods were fed with *Chilomonas paramecium* infusion that was cultured with wheat seeds (50seeds/L) in 1-L plastic jars at 27 °C for seven days or longer. The chemical oxygen demand (COD) of infusion always exceeded 100 mg/L according to our measurements using commercial kits (Kyoritsu Chemical-Check Lab., Corp., Tokyo, Japan).

### 2.2. Mosquito Colonies

A laboratory colony was established with *Ae. aegypti* females collected from Ho Chi Minh City, Vietnam, in 2005. Adult mosquitoes that emerged were maintained at 27 ± 1 °C and 70% relative humidity under 14L/10D (14 h light - 10 h dark) photoperiod conditions. They were fed with a 3% sucrose solution and allowed to suck blood from an anesthetized mouse once a week. Newly hatched *Ae. aegypti* first instar larvae (within 2 hours after hatching) were used in the experiments.

### 2.3. Experimental Design

Experiments were carried out under eight conditions (two larval densities, with or without copepod, and two nutritious conditions) at three temperatures (20, 27, and 32 ± 1 °C) under 14L/10 D photoperiod in incubators (Biotron LPH–220/350S, Nihon-ika Corp, Osaka, Japan). We measured water temperature in experimental cup without animals and foods placed in the incubators using StowAway Tidbit data loggers (Onset Computer Corp, MA, USA), and we confirmed that water temperature fluctuated within the range of ± 1 °C. The population performance of *Ae. aegypti* was evaluated by larval mortality, the duration of the first-instar and whole-larval periods, and pupal dry weight [23]. The performance of *M. aspericornis* was evaluated by the number of offspring produced and the development time of offspring from egg to adulthood. A treatment comprises of 10 or 50 newly hatched *Ae. aegypti* larvae with or without a primiparous *M. aspericornis* female in a 200-mL plastic cup (8 cm in diameter; 4.5 cm in height) containing 100 mL of “nutrient rich” or “nutrient poor” water. To prepare *Ae. aegypti* larvae, dried eggs were soaked in water for 2 hours, and then newly hatched larvae were collected. The nutrition medium was *Chilomonas paramecium* infusion. The COD level of “nutrient rich” water was adjusted to 20 mg/L by diluting the infusion, and 5 mL of food medium was added daily while the COD level of “nutrient poor” water was adjusted to 4 mg/L and one mL of food medium added daily. Thirty replicates were prepared per treatment.

### 2.4. Rational of Experimental Settings

Temperatures of 20, 27 and 32 °C were chosen to cover the range of the annual minimum and maximum temperatures of Ho Chi Minh, where the mosquito strain was collected. The COD of water in experimental cups was determined with reference to COD in mosquito habitats in Ho Chi Minh; i.e., COD in field containers in which *Aedes* larvae occurred was approximately 20 mg/L as a mode (TVP, unpublished data). All treatments were checked daily to record the number of surviving mosquito larvae and growth of copepods. When mosquito pupae were observed, they were collected and dipped in 60 °C hot water to kill. Then, they were placed on paper using a pipette to absorb moisture, transferred to a 96-well plastic plate to dry at 32 °C for 48 hours, and measured for dry weight using a digital balance (MX5-Mettler Toledo Laboratory Weight, Greifensee, Switzerland). Observations were terminated when all mosquitoes had pupated or died.

### 2.5. Statistical Analyses

The effects of temperature, nutrition conditions, initial larval density and the copepod presence on mosquito performance was analyzed by generalized linear model (GLM). The analysis on mortality was performed with the logit link function and the binomial distribution error, and the analysis on the larval development time and mosquito pupal weight was with the identity link function and the normal distribution error. The effects of temperature, nutrition, initial larval density on copepod performance was also analyzed using GLM with the identity link function and the normal distribution error. For explanatory variables, temperature, nutrition, initial larval density and the copepod presence and interaction terms of these parameters were used in the analysis of mosquito performance, while the same parameters, except for the copepod presence, were applied in the analysis of copepod performance. All statistical analyses were performed using JMP version 11.2.1 (SAS Institute, Cary, NC, USA).

## 3. Results

### 3.1. Effects of Copepod Presence on the Larval Performance of Ae. aegypti

The GLM showed that all factors (i.e., temperature, nutrition condition, initial density of *Ae. aegypti* larvae and copepod presence) significantly affected larval mortality and developmental time and pupal body weight of *Ae. aegypti* (Table 1, *p* < 0.001). The mortality of first instars was most influenced by the presence of copepod (Table 1), while no mortality was observed for them irrespective of rearing conditions when copepod was absent. Significant effects were also observed on the interactions of copepod presence with temperature, nutrition conditions, and the initial density of larvae, indicating that the effect of copepod presence varies depending on conditions (Table 1, Figure 1A). The presence of copepod also had significant effects on mortality during the whole larval period, but the effects were much smaller than the effects on the mortality of first instars (Table 1).

Mortality during whole larval period was significantly higher, and larval development time was longer when rearing temperature was lower, nutrition conditions were poorer and the initial larval density was higher (Table 1, Figure 1A,B). On the other hand, pupal dry weight was generally significantly lighter when rearing temperature was higher, nutrition conditions were poorer and the initial larval density was higher, but temperature did not have significant effects on pupal weight when reared under nutrition-rich and low-larval-density conditions (Table 1; Figure 1C). These results suggest that *Ae. aegypti* larvae were subjected to intraspecific competition under the present experimental conditions. 

The presence of copepod increased mortality of *Ae. aegypti* larvae (Figure 1A), but larval period was shortened (Figure 1B) and pupal dry weight increased (Figure 1C). These results suggest that the intraspecific competition of *Ae. aegypti* larvae was relaxed to some extent due to predation by copepods. Under the presence of copepods, the mortality of *Ae. aegypti* larvae (i.e., predatory efficiency of copepods) was generally higher at a lower temperature, under nutrition-poor conditions and at a low initial density of *Ae. aegypti* larvae (Table 1, Figure 1A). 

### 3.2. Effects of Temperature, Nutrition Conditions and Larval Density of Ae. aegypti on Copepod Performance

The GLM showed that all factors (i.e., temperature, nutrition conditions and the initial density of *Ae. aegypti* larvae) significantly affected the production and developmental time of copepod offspring (Table 2). The number of offspring was generally significantly larger and developmental time was shorter at a higher temperature, under nutrition-rich conditions and a lower initial density of *Ae. aegypti* larvae (Table 3).

## 4. Discussion

After the findings that cyclopoid copepods substantially feed on mosquito larvae in the field [6,16,24], several studies have proved that copepods are an effective agent to control mosquito populations in northern and central Vietnam [2,3,4,5,7], but the utility of copepods was not always obvious [11]. Here, we studied how a copepod species, *Mesocyclops aspericornis*, affects the performance of a prey mosquito, *Ae. aegypti*, under different temperatures, nutrition and larval-density conditions to understand what affects the utility of copepods. We found that predation by copepods reduced mosquito density and thereby their interspecific competition was relaxed, resulting in the faster development and larger body size of survivors. The results are consistent with a previous study on *Mesocyclops pehpeiensis* and *Ae. albopictus* [15]. Thus, predation may increase survivorship and reproductive output of the survivors of prey species. 

Under the presence of *M*. *aspericornis*, the mortality of *Ae. aegypti* larvae (i.e., predatory efficiency of copepods) was generally reduced at higher temperatures, under nutrition-rich conditions and at a high initial density of *Ae. aegypti* larvae. This seems to be related to the prey-size selection of copepods and to the difference in their temperature response. Copepods preferentially feed on small mosquito larvae [6,14], possibly because larger mosquito larvae can resist copepod attack. At higher temperatures and under nutrition-rich conditions, *Ae. aegypti* larvae increase body size more rapidly and could escape from copepod attack. This result may explain why the application of copepods was not successful in a hot location (i.e., southern Vietnam). It may also explain the results of copepod application in Laos where *Ae. aegypti* was eliminated from wells but not from water containers [11,25]; i.e., water temperature may be higher or food conditions were better in water containers than in wells. In such locations or water types, it will be effective to apply additional means on top of the sustainable biological control. For example, the application of insecticides such as *Bacillus thuringiensis israelensis* (Bti), permethrin, methoprene, or pyriproxyfen may be effective, because these kill mosquito larvae but do not harm copepods [7,26]. Furthermore, it was found that a low doses of mosquitocidal nanoparticles help to boost the control of *Anopheles* and *Aedes* populations under copepod-based control programs [27]. 

As a new method of controlling mosquitoes—a mass release of genetically modified mosquitoes—is becoming applicable in addition to the conventional means. Yakob et al. performed a mathematical simulation of three control methods, a mass release of infertile males, Wolbachia-infected mosquitoes and genetically modified mosquitoes carrying lethal genes, and pointed out that the former two may enhance the survival of adult mosquitoes by reducing intraspecific competition of mosquitoes [28]. Moreover, it should be noted that neither a mass release of genetically modified mosquitoes nor a mass release of infertile mosquitoes is sustainable long-term control methods, unlike the copepod method, and require large mosquito breeding factories for large-scale release. Each of the control measures has its advantages and disadvantages in terms of economic efficiency, immediate effect, development of drug resistance and sustainability. In order to fully utilize the good points of each control measure, it is necessary to understand the characteristics of each method. In addition to sustainable long-term control by copepods, the preparation of the short-term use of other methods having immediate effects will be effective in cases where the population growth of dengue vectors is high.

## 5. Conclusions

Our study demonstrates that copepod predation could reduce intraspecific competition among *Aedes* larvae and result in a faster development and larger pupal weight of survivors. In addition, high temperature, nutrition-rich conditions and a high larval density of *Aedes* reduced the predatory efficiency of *M. aspericornis*, probably because *Aedes* larvae increased body size more rapidly and were able to escape from copepod attack. The effect of a high density of mosquito larvae is to hamper copepod growth through interspecific competition for food. These results suggest that the utility of copepods as a mosquito-controlling agent could be reduced under higher temperature, organic rich water, and higher densities of mosquitoes.

## Figures and Tables

**Figure 1 insects-11-00307-f001:**
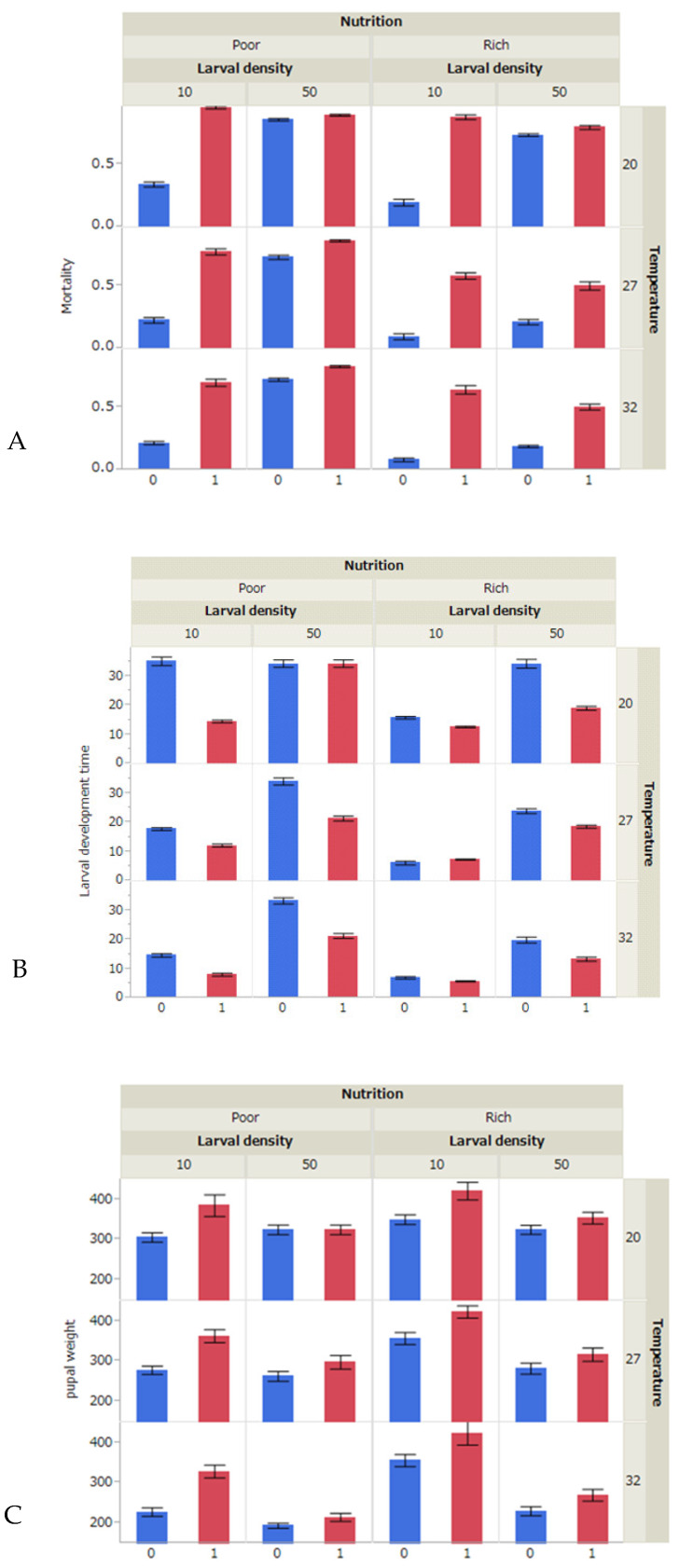
Mortality, larval duration and pupal dry weight of *Aedes ageypti*. Mortality (rate: mean±SD) during the whole larval period (**A**) when reared under different temperatures (20, 27 or 32 °C), nutrition (Rich or Poor) and initial larval-density (10 or 50 larvae) conditions with (1) or without copepod (0). Larval duration (days: mean ± SD) of the whole larval period (**B**) and pupal dry weight (µg: mean±SD) (**C**). The number of replications was 30 for each treatment.

**Table 1 insects-11-00307-t001:** Summary of generalized linear model (GLM) analysis on larval mortality during the first instar and whole larval periods, larval duration, and pupal dry weight when reared under different temperatures (20, 27 or 32 °C), nutrition (Rich or Poor) and initial larval-density (10 or 50 larvae) conditions with (Cope) or without (Control) copepods. NS means not significant at the level of *p* > 0.1.

Parameter	Response Variables of *Ae. aegypti*							
	Mortality				Larval Developing Duration		pupal Dry Weight	
	First Instar Period		Whole Larval Period					
	Likelihood Ratio χ^2^	*p*	Likelihood Ratio χ^2^	*p*	Likelihood Ratio χ^2^	*p*	Likelihood Ratio χ^2^	*p*
Copepod	1630	<0.0001	63	<0.0001	76	<0.0001	8	0.0147
Temperature	31	<0.0001	14	0.0002	228	<0.0001	29	<0.0001
Larval density	44	<0.0001	19	<0.0001	173	<0.0001	9	0.0034
Nutrition	33	<0.0001	30	<0.0001	122	<0.0001	17	<0.0001
Cope ×Temp	50	<0.0001	0	NS	14	0.0008	1	NS
Cope × Larv	63	<0.0001	21	<0.0001	15	0.0001	18	<0.0001
Cope × Nutr	26	<0.0001	1	NS	26	<0.0001	0	NS
Temp × Larv	2	NS	0	NS	12	0.0023	26	<0.0001
Temp × Nutr	5	0.024	2	NS	6	0.0449	12	0.0024
Larv × Nutr	0	NS	2	NS	0	NS	16	<0.0001

**Table 2 insects-11-00307-t002:** Summary of GLM analysis on the number and developmental duration (egg to adult) of *M. aspericornis* offspring.

Parameter	Response Variables of *M. aspericornis*			
	Number of Offspring		Developmental Duration	
	Likelihood Ratio χ^2^	*p*	Likelihood Ratio χ^2^	*p*
Temperature	19	<0.0001	130	<0.0001
Larval density	23	<0.0001	236	<0.0001
Nutrition	0	NS	100	<0.0001
Temp × Larv	17	0.0002	120	<0.0001
Temp × Nutr	3	NS	77	<0.0001
Larv × Nutr	0	NS	47	<0.0001

**Table 3 insects-11-00307-t003:** Production of *M. aspericornis* offspring (N: mean ± SD) and their development time (days: mean ± SD) when reared under different temperatures (20, 27 or 32 °C), nutrition (Rich or Poor) and initial mosquito larval-density (10 or 50 larvae) conditions. The different letters indicate significant difference.

Food	10 Larvae			50 Larvae		
	20 °C	27 °C	32 °C	20 °C	27 °C	32 °C
**Copepod offspring abundance**						
Rich	15.7 ± 9.0 ^BCD^	79.9 ± 34.6 ^A^	70.7 ± 33.7 ^A^	9.6 ± 8.5 ^CD^	16.7 ± 8.6 ^BCD^	41.8 ± 18.3 ^B^
Poor	12.4 ± 4.5 ^BCD^	29.5 ± 11.4 ^BC^	45.9 ± 15.8 ^ABC^	4.5 ± 2.3 ^D^	7.9 ± 6.5 ^CD^	8.8 ± 4.8 ^CD^
**Developmental time (day)**						
Rich	15.8 ± 1.4 ^de^	9.4 ± 1.1 ^g^	5.7 ±1.0 ^h^	21.9 ± 3.5 ^b^	15.1 ± 3.4 ^ef^	8.5 ± 1.4 ^g^
Poor	18.9 ± 1.7 ^c^	14.1 ± 3.2 ^ef^	9.2 ± 2.4 ^g^	52.3 ± 11.2 ^a^	18.2 ± 2.7 ^cd^	12.7 ± 2.2 ^f^

A–D, a–h: The different letters indicate significant difference.

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
