# Peer review of "Climate Change May Restrict the Predation Efficiency of Mesocyclops aspericornis (Copepoda: Cyclopidae) on Aedes aegypti (Diptera: Culicidae) Larvae"

_insects, 2020, doi:10.3390/insects11050307_

Round 1

Reviewer 1 Report

Please, see the attached document for detailed comments, suggestions and questions.

Overall, I think that the work can be improved. the figures are quite difficult to read and interpret (the legends and lines are too similar among samples - try using box plots or a different color pattern, or splitting the graphs in other panels.

It is essential that some speculation on the discussion to be corrected or removed. and a literature update in some aspects.

Some of this speculation is related to the offspring survivor, which could be detected having control using copepods without mosquito larvae. Please consider including this information properly, or a reference.

Author Response

We appreciate your precious time and careful revision and constructive discussion.

We change the ms accordingly.

Our reply to your respective comment is written in red (in the attached file).

page :1

L14: since no specific and effective

We follow the change.

page :2

L48-50. Sentence is confusing and no clear what you mean.

We rewrite as “Furthermore, copepods are predators for younger mosquito larvae, but are competitors for later-stage larvae competing for food”.

L55-56. Are there any information on genomic selection and adaptation to avoid/reduce predation?

No evolutionary adaptations of mosquito larvae to copepod predation have been reported to our knowledge.

L57

-this statement is possible only if there is enough food for development. otherwise, only some individuals will reach adulthood under

We rewrite “provided that food is not limited and it does not exceed harmful upper limits”

L74. Do you know from how many females this colony was established? was there any genomic background refreshment using material recent collected from the field?

Since we collected eggs, it is unknown how many female were involved in the colony. After the original collection, wild individual were not added.

L76. Usually is a 10% sucrose solution, why the decision to reduce sugar concentration for adult feeding?

Our laboratory keeps all mosquitoes with 3% sucrose water as a routine. I asked the reason to the previous staff since I have also been wondering it. There was a reason but was not biological one. I remember your comment for our further study.

L76. Was there any twilight period?

There was no twilight period.

L77. Which method was used for hatching? did the L1 had access to food during this 2 hours? how the L1 larvae were manipulated or offered to the M. aspericornis?

We harvested the first instar larvae that hatched from dried eggs those were soaked in water after 2 hours. There was no food in the water.

L77. Is there any ethics committee aware of this procedure? Was the mouse anesthetized?

Animal experiments are submitted and being reviewed by an ethic committee organized by the institute. Mouse is anesthetized for 30 to 50 min by intramuscular injection of 80μl (30μl/ 10g of body weight of mouse) of cocktails (ketalar 2μl, xylazine 500μl, saline 500μl) and used. We rewrite as an anesthetized mouse.

L80. Is there a constant number of M. aspericornis offspring? was it constant the number of copepods throughout the samples?

There was not constant number of M. aspericornis offspring. There was constant number of copepod.

We originally introduce one M. aspericornis female in the copepod treatment groups. We terminated the rearing experiment according to the rule “Observations were terminated when all mosquito had pupated or died or when all copepod became adults or died”. In reality, the experiment ended when all mosquitoes die or pupate.

Accoringly, we rewrote the part as “Observations were terminated when all mosquito had pupated or died”.

All experimental systems of copepod-treatments consist of a female copepod and immature stage of copepod if they were born, meaning that there was only one predatory copepod. Since immature copepods are not predatious.

L83. Were they alive? how could you control its spread?

They are alive. We add wheats and water periodically to keep their colony constant in the rearing bottles. Their concentration adjusted by measuring chemical oxygen demand before addition as explained in the text.

L85. Please describe better the experimental groups. Things are confusing with these terms and inclusion of food medium.

The feeding regime is mixed with the food medium preparation and it's really hard to understand it.

We rewrote the part to avoid confusion.

L85.

,,,also it is not clear about your controls. Did you set up a negative control for M. aspericornis without mosquito L1?

There is one type of control. This experiment is not a competition experiment between two organisms, but to investigate whether the presence of the copepod can suppress the growth of Aedes aegypti. We are not interested in measuring the population growth of copepod in the absence of mosquitoes.

L88. Is the mode selected due to the great standard deviation found among the samples? Is it possible to give some more information about the sampling of this study? Such as breeding site type, collection frequency, geographical location, other relevant information.

We used the mode instead of the average because the range of variations in water pollution was large, but most of the measurements were around 20mg/L. Unlike Culex quinquefasciatus, Aedes aegypti were not found in very organic rich water. We plan to write a separate paper on this field survey, and we would like to avoid detailed description in this paper.

L90. Why those temperatures were selected? is there a specific need for each organism? Once both species colonies were kept at 27 degrees as mentioned on 2.1 and 2.2. and there were no light difference for the experiment.

The three temperatures were set based on the annual minimum and maximum temperatures of Ho Chi Minh, where mosquitoes were originally collected. We added an explanation in the text. In addition, 27 ° C is the temperature at which we keep mosquitoes, so that we could reduce the number of incubators needed. There were only three incubators of the same size that could be used. For example, it was impossible to control the number of protozoa used as food. We thought that it was more important to carry out experiments under different conditions at the same time than to carry out different experiments in sequence. Regarding the relationship between the temperature and the day length condition, which the reviewer may be considering the experimental setting at different latitudes, but we thought that global warming means the temperature rises at the same latitude.

L91. Does it means that you had a total of 30 M. aspericornis being evaluated individually?

The cup containing copepod had 30 repetitions under different conditions, and 30 copepoda were used individually. The group without copepod was measured with the same number of breeding containers.

page :3

L93. Please provide more information. Is there a reference describing the method? was only the pupae sieved and placed in a glass Petri dish for drying?

There is no reference for the method. We rewrote the relevant part as “We dropped newly pupated individuals in 60 ° C hot water to kill, picked up them using a pipette, transferred them on paper to absorb surface moisture , transferred to a 96-well plate, and dried at 32 ° C for 48 hours.”

Table 1.

Wrong symbol for Celsius degrees.

It is corrected.

page :4

Table 2

Consistent table formatting.

We would appreciate it if the reviewer 1 could clarify what you are pointing out. If you mean that the number of samples is not shown in Table 1, the number of samples in Table 1 is 10 individuals x 30 containers or 50 individuals x 30 containers, that is, 300 individuals or 1500 individuals. We have inserted the numbers in Table 1.

L140.

Is there any physical evidence that predatory activity is the mortality mean factor?

It is common to see the larval carcasses killed in a container, as copepod kills a greater number than they eat. However, it was not determined whether individual cause of death was due to predators.

page :5

Figure 1. A and B figure reference in the figure is missing.

Thank you. We changed the figures in the revision.

page :7

L 190. This sounds speculative and predatory activity indirectly increase body size. Have you considered keep the mosquitoes which survive to adulthood and check if their behavior could shift the findings along the generations?

We corrected “the survivors” instead of “the following generations”. “Thus, predation can increase survivorship and reproductive output of the survivors of prey species.”

L192. Completely speculative. The reference is from a complete different species, which has no role on dengue transmission. Please provide a more suitable reference.

Please consider reading:

PMID: 9673914

DOI:10.1111/j.1365-2915.2007.00694.x

DOI: 10.1093/jmedent/47.5.778

doi: 10.1371/journal.pone.0059933

doi:10.1073/pnas.1101377108

L195. speculation without any reference, please consider changing

Thank you for the information. We read the references and deleted the relevant part.

L203. And how about increasing the copepods number on those breeding sites? And how well distributed are they in the environment?

This reference is a kind of social science paper, not a survey of copepod, and no details are reported. We just understand that copepod population was extinct frequently in the container in which they were introduced in their fields. No further information is available. TVP, who is from Vietnam, said that they conducted and reported many successful projects of biocontrol using copepod, but in reality it was not very successful in the South. After discussing why, we got the hypothesis that the copepod may slow down in nutrition-rich or warmer water bodies, and we tried to confirm it in this study.

L.208. Vector control cannot rely only in one method, and this is well known.

Please consider reading:

doi: 10.1371/currents.outbreaks.45deb8e03a438c4d088afb4fafae8747

doi:10.1016/S0140-6736(11)60246-8

DOI: https://doi.org/10.4269/ajtmh.2008.78.70

doi: 10.3201/10.3201/eid1206.051210

doi: 10.1371/journal.pntd.0004551

Thank you for the information. We rewrote the relevant part referring to one of them.

L.210. It's known about selection of resistant mosquito population and the impact on other organisms. In addition most insecticides cannot be applied on water for human use/consumption.

Thank you for the information. We rewrote the relevant part referring to those references.

page :8

  1. 216.

That's why a control not using the mosquito larvae is crucial to properly evaluate the conditions of the experiment.

We agree the opinion. We have reported copepod reproductive traits in elsewhere. 

PMID: 18437816, https://doi.org/10.2987/5672.1

PMID:16506581

PMID:12674538

PMID:12243229

L.229.

not only that, copepods can also have a different diet from mosquitoes and they stop competing, and copepods then are not depending on L1 larvae to survive. Do you think that other factors related to the environment can be also affecting the copepods? the temperature and bacteria development, removing oxigen from water could be one factor? The metabolic result secreted by the larvae can also play an important role, specially on higher densities.

Thank you for the interesting comment. Unfortunately we don't have applicable data to give you the right answer. What we can see from this result is that the mortality rate of mosquitoes decreases as the temperature increases. Both temperature and nutrition have a negative relationship with the amount of dissolved oxygen, but in the analysis, nutrition did not affect the copepod performance as much as temperature. Therefore, at least based on the results, we think that the temperature itself rather than the dissolved oxygen controls the growth of copepod. In the three temperature ranges applied, the growth of copepod decreased at higher temperatures, while the mosquitoes became better at higher temperatures. We report that aegypti growth was improved up to 30 degrees. https://doi.org/10.1111/j.1365-2915.2011.00971.x

What we can say in this work is that this is because the optimum temperature of the two organisms is higher in aegypti, and it is predicted that global warming will make it difficult for biological control of copepod.

  1. 231. The paper focus is in reducing disease transmission by vector control using copepods. Besides being already in use in some places, how could the finding improve the application o this method? and how to propagate this control method to other places and what would be the key parameters for its implementation?

Thank you for your valuable feedback. As mentioned in the introduction, copepod application has actually been successful in controlling mosquito populations in the field. Currently, it has been proposed to mass-release genetically modified mosquitoes and Wolbachia-infected mosquitoes, but we think that verification of them in the field is not yet sufficient. In addition, GMM are, in principle, the same as the conventional mass release of infertile male flies, which requires huge incubators for mass-release and is not sustainable. Biocontrol using copepod is a cheap, special equipment-free and sustainable method. We think that it is important to clarify in what environmental conditions the effects can be expected or not, because the control by copepod may not always succeed due to the effects of global warming.

Reviewer 2 Report

Introduction

You should update your stat-of-the art with recent references. Provide more details why you choose this vector (e.g and not Aedes albopictus or Culex pipiens) a to conduct your studies. Your introduction is adequate but you should focus more to your scope (and not to a non-successful application in Vietnam).

Material and methods

Mosquito colonies. Give more details for your egg hatching protocol. This is important to understand how you managed to collect newly hatched first instar larvae within 2 hours (after hatching). Furthermore, mosquitoes collected in the field can be preserved for purposes of identification and screening to see whether or not they harbour any arboviruses. Please comment.

Experimental design. Explain in detail why you select these 3 temperatures. It is important to associate them (the chosen temperature) with the biology of the vector and the predator. The correct selection of temperatures is critical point for vectors. For example, in Table1, “mortality during whole larval period” was decreased in higher temperatures. This phenomenon may create controversy results since the number of samples measured were significant higher in all treatments at 32 degrees (table 2). It is known that Ae. aegypti colonies performed better in higher temperatures than Ae. albopictus. Therefore, the larval duration was shorter at 32 degrees.  

Finally, the most critical point that authors should consider for these experiments is the “real temperature” inside the water. This means that the temperature of the incubator is never the same with the water temperature. Advice for authors is that treatment at e.g. 20 degrees generally requires that the thermostat be set at higher temperature. Thus, trial must be conducted in several type of containers (different shape, material etc) at selected temperature for calibration purposes.  

Recommendation

The study appears to be technically well executed and t merit to be published to Insects after major revision.

Author Response

Dear Reviewer 2.

Thank you very much for your time and careful reading of our manuscript.

You should update your stat-of-the art with recent references. Provide more details why you choose this vector (e.g and not Aedes albopictus or Culex pipiens) a to conduct your studies. Your introduction is adequate but you should focus more to your scope (and not to a non-successful application in Vietnam).

Our reply: We changed the statistical analysis. We chose Aedes since copepod application has been conducted for control of dengue fever. Why we did not use Aedes albopictus is due to the handling problem. Eggs of Ae. albopictus do not hatch immediately so it was harder to do this study using the species. This is not scientific reason and we did not mention it in the text.  

Material and methods

Mosquito colonies. Give more details for your egg hatching protocol. This is important to understand how you managed to collect newly hatched first instar larvae within 2 hours (after hatching). Furthermore, mosquitoes collected in the field can be preserved for purposes of identification and screening to see whether or not they harbour any arboviruses. Please comment.

Our reply: We add more information in the method.

L. 87. We harvested the first instar larvae that hatched from dried eggs those were soaked in water after 2 hours.

Experimental design. Explain in detail why you select these 3 temperatures. It is important to associate them (the chosen temperature) with the biology of the vector and the predator. The correct selection of temperatures is critical point for vectors. For example, in Table1, “mortality during whole larval period” was decreased in higher temperatures. This phenomenon may create controversy results since the number of samples measured were significant higher in all treatments at 32 degrees (table 2). It is known that Ae. aegypti colonies performed better in higher temperatures than Ae. albopictus. Therefore, the larval duration was shorter at 32 degrees.  

Our reply: We add Rational of experimental settings. in the text line 92. 

Finally, the most critical point that authors should consider for these experiments is the “real temperature” inside the water. This means that the temperature of the incubator is never the same with the water temperature. Advice for authors is that treatment at e.g. 20 degrees generally requires that the thermostat be set at higher temperature. Thus, trial must be conducted in several type of containers (different shape, material etc) at selected temperature for calibration purposes.  

Our reply: In our incubator, the water temperature coincide with the setting temperature. We sometimes measured water temperatures using HOBO tidbit water proof loggers.

Round 2

Reviewer 1 Report

I appreciate the efforts to update the manuscript and reply to my questions. I have no further comments.

Author Response

Dear, Reviewer 1.

Thank you very much for your instructions.

We are very grateful for your constructive feedback, based on your wealth of knowledge.

Sincerely yours

Nobuko Tuno